# The role of race and scientific trust on support for COVID-19 social distancing measures in the United States

**Sara Kazemian**[1]☉, **Sam Fuller**[1]☉*, **Carlos Algara**[2]☉

**1** Department of Political Science, University of California, Davis, Davis, California, United States of America, **2** Division of Politics & Economics, Claremont Graduate University, Claremont, California, United States of America

☉ These authors contributed equally to this work.
* sjfuller@ucdavis.edu

**Data Availability Statement:** The datasets and replication materials are available on Harvard Dataverse (https://doi.org/10.7910/DVN/0RMZHK).

**Funding:** The authors received no specific funding for this work.

## Abstract

Pundits and academics across disciplines note that the human toll brought forth by the novel coronavirus (COVID-19) pandemic in the United States (U.S.) is fundamentally unequal for communities of color. Standing literature on public health posits that one of the chief predictors of racial disparity in health outcomes is a lack of institutional trust among minority communities. Furthermore, in our own county-level analysis from the U.S., we find that counties with higher percentages of Black and Hispanic residents have had vastly higher cumulative deaths from COVID-19. In light of this standing literature and our own analysis, it is critical to better understand how to mitigate or prevent these unequal outcomes for any future pandemic or public health emergency. Therefore, we assess the claim that raising institutional trust, primarily scientific trust, is key to mitigating these racial inequities. Leveraging a new, pre-pandemic measure of scientific trust, we find that trust in science, unlike trust in politicians or the media, significantly raises support for COVID-19 social distancing policies across racial lines. Our findings suggest that increasing scientific trust is essential to garnering support for public health policies that lessen the severity of the current, and potentially a future, pandemic.

## Introduction

With more than 3 million dead and nearly 150 million infected across the world, as of this writing, the COVID-19 pandemic is an unprecedented international health crisis wreaking a devastating human toll. Moreover, racial and ethnic minorities in the United States (U.S.) have been disproportionately impacted by COVID-19. Studies show a higher incidence of COVID-19 related mortality among racial/ethnic minorities generally [1], and among those who are essential workers specifically [2]. For example, while Black Americans represent 13 percent of the U.S. population, they account for 24 percent of the COVID-19 related fatalities [3]. Furthermore, in the District of Columbia (D.C.), Black Americans reflect a 50 percent share of the population, but account for 75 percent of COVID-19 deaths [3]. Encouragingly, with the

**Competing interests:** The authors have declared that no competing interests exist.

ongoing mass vaccination of Americans, there is a significant opportunity to reduce these massive racial disparities. However, there is a significant threat that these disparities may persist or even worsen given the extensive research on how Black Americans tend to be much less trusting of medicine, particularly immunization programs, than White Americans and slightly less trusting than Hispanic Americans [3].

Given that COVID-19's spread is heavily determined by interpersonal interactions, restrictions meant to increase social distancing have been implemented widely, with varying degrees of success and compliance, by local, state, and national governments [4]. Support for and compliance with these policies, evidenced by widespread anti-"lockdown" and anti-mask protests/ behaviors and surveys conducted by Pew Research Center [5], is highly variable, with many heavily in favor and nearly as many heavily opposed. Recent work has attempted to parse out the likely determinants of support and compliance, examining influences such as gender, partisanship, and scientific knowledge and trust [6, 7]; race/ethnicity, conspiracy theory beliefs, and COVID-19 knowledge [8]; and local coronavirus incidence and threat perceptions [9].

The importance of this standing work evaluating the determinants of support for government policies aimed at stopping the spread of COVID-19 is underscored by the classic representational idea that government action is predicated on the support of the mass public. Indeed, stopping the spread of the COVID-19 pandemic represents a stark collective action problem, with government policies being critical in ensuring social distancing occurs in society rather than relying on potentially unreliable compliance by the mass public with public health guidelines [4, 10]. Given that implementation of policies by election-seeking representatives is conditional on the degree of support found in the mass public, it is critical that scholars identify the dynamics of public support for government social distancing policies. This is evidenced not only by public health guidance, but also previous work that finds that in the absence of government social distancing policies, much higher rates of COVID-19 infections and deaths would have occurred [4, 10]. In sum, it is critical that scholars identify potential mechanisms that increase public support for these policies and, in turn, the likelihood of government implementation by office-seeking elected elites.

Recently, in the midst of the pandemic, scholars have turned their attention to the relationship between political trust, scientific trust, and COVID-19 outcomes. Thus far, there is mixed evidence on the relationship between public trust in science and the COVID-19 pandemic. Indeed, some document that the pandemic has increased the public's trust in science [11], and impacted the extent to which people trust institutions [12]. Conversely, others report a negative relationship and caution that the pandemic has eroded public trust in science and reduced the willingness to get a vaccine [13]. Along these lines, Thaker [14] underscores the importance of scientific trust and argues that trusted scientific experts are crucial in increasing vaccination rates. However, it is unclear how trust in science, politicians, and the media influence support for public health recommendations such as shelter in place policies. Previous research finds that when people distrust politicians they may also distrust the policies that the government formulates [15]. Indeed, (dis)trust in politicians can influence support for specific policies ranging from $CO_2$ taxes [16] to redistributive policy preferences [15]. Furthermore, research conducted explicitly about pandemic policy responses has found that (dis)trust in the government/politicians is a strong determinant in support of, or opposition to, shelter in place policies [17, 18].

Devine et al. [19] review the literature on trust during the COVID-19 pandemic and argue that trust is a complicated construct that not only shapes COVID-19 related policies, but is also shaped by the pandemic itself. The authors highlight that trust improves compliance with social distancing policies and helps mitigate the spread of COVID-19. Nadelson et al. [20] similarly emphasize that scientific trust is a multi-faceted construct and argue for a contextual

approach in understanding scientific trust. For example, the authors contend that public trust in vaccines is likely influenced by both emotional and historical contexts. Importantly, public trust in science tends to cut across partisan and racial lines. For example, Democrats are more likely than Republicans to believe that scientists act in the best interest of society [21, 22]. Black Americans—who predominately identify with the Democratic party—however, are still more skeptical of science than White Americans [3]. This trust gap between White and Black Americans further fuels our investigation of the role of scientific trust on support for COVID-19 policies.

Finally, literature on the influence of traditional media (newspapers, television, radio) on health-related behaviors is well documented, yet there is mixed and scant evidence on how media usage and trust varies across different races and ethnicities [23]. Pandemic-related research has found that individuals' consumption of and trust in traditional media influenced the adoption of preventative behaviors and vaccination intention during the 2009 H1N1 pandemic [24, 25] and the current pandemic [26]. Importantly, the most recent literature on media's influence has found that its effects are largely divided along partisan lines—that is, if you consume conservative media in the United States you are less likely to practice social distancing [26]. Alternatively, if you consume liberal media, you are more likely to practice social distancing [26].

The article proceeds as follows: First, we note the existing literature that highlights the link between scientific trust and support for social distancing policies and the racial disparities in scientific trust and resulting behaviors. Second, we reinforce previous findings on the racial disparities in COVID-19's human toll in the U.S. using a county-level, high-dimensional regression. Third, given previous findings and the pandemic's unequal impact on communities of color in the United States, we investigate the interactive influence of race/ethnicity and trust in science, politicians, and the media in determining support for social distancing policies/restrictions. We find that scientific trust influences support for both individual social distancing policies (such as restricting large gatherings) and two composite measures of all polled policies, and that these effects are particularly strong among Black Americans. Overall, we contend that increasing scientific trust among Black Americans is likely a very important and effective pathway for increasing support for social distancing policies and thus decreasing the unequal effects of COVID-19 and future pandemics on communities of color.

## Scientific trust & COVID-19 social distancing policy support

Trust, be it political or scientific, is an important ingredient in any functioning society. News media and academic researchers alike often cite public trust in science as an important requirement for pro-social behavior and adherence to policy recommendations [27]. Trust can be defined as the "the willingness of a person, group or community to defer to or tolerate, without fear, the judgments or actions of another person in institutions that directly affect one's actions on welfare" [28]. In other words, trust is the decision to accept vulnerability and give another person "benefit of the doubt" [29].

Scientific trust is key in understanding how ordinary people reach conclusions about public health. When the public trusts scientists, they place confidence in the scientific community to provide expert knowledge on important public policies such as public health, education, energy consumption, and climate change. Critically, public trust in science is especially important when the public has a poor understanding of the risks associated with a new technology or a public health crisis, like COVID-19. Thus, if the average citizen is uninformed about a new technology (like a vaccine), or new public health recommendations (like social distancing), they may rely on scientists to inform their opinions [30]. Unsurprisingly, the prevailing

consensus is that scientific trust underpins successful immunization programs [31], environmental policies, as well as support for social distancing [32]. As communities around the country combat COVID-19 and make plans to reopen their economies, policymakers will have to rely on public support for social distancing, mask mandates, and widespread testing.

## Scientific trust & racial disparities in public health

Although it is obvious why trust in science is important, it is less obvious why some groups have higher scientific trust than others. Even less clear is whether the relationship between scientific trust and support for scientific policies is moderated by a person's race or ethnicity. We draw on literature from immunization programs in the United States and investigate whether scientific trust's influence on support for social distancing policies is potentially moderated by race/ethnicity.

Research on immunization programs has consistently documented a racial trust gap between Black Americans and White Americans and support for the yearly influenza vaccine [31, 33, 34]. Much of the previous literature attributes this gap to historical discrimination of Black Americans in the medical community [35]. The Tuskegee syphilis study is the clearest and most well known example of why Black Americans may distrust medicine, physicians, and medical recommendations generally. The study, which was intended to last between 6 to 8 months, recruited 400 Black American men with syphilis who had not yet received any treatment. Despite the designated time frame, the study ran for 40 years, even though penicillin became available during the duration of the experiment [36]. The study's use of deception, as well as mistreatment of participants, is a key reason why Black Americans mistrust science and medicine. Today, research indicates that knowledge of this history influences modern day perceptions about the medical community in a way that increases African Americans' mistrust of medical professionals and expectations of dishonesty from scientists [37].

To examine this relationship further, Scharff et al. [38] interview 11 focus groups and find that mistrust in medicine originates from unethical medical research and continues to have lasting effects in the African American community today. Similarly, Freimuth et al. [31] document a racial immunization gap in the influenza vaccine: 53.4% of White participants reported getting a vaccine, compared to 44.4% of African Americans. Importantly, the authors reveal that "the effect of racial consciousness was a negative predictor for both [White and Black] groups but was only significant for African Americans." Put simply, when Black adults think about race in a healthcare setting, they are less likely to trust the influenza vaccine. However, racial consciousness has no effect on White Americans. Thus, racial factors such as historical discrimination and racial consciousness have clear and disproportionate effects across race. Beyond these historical factors, present-day examples of medical mistrust have focused on either personal experiences [39], or vicarious experiences [40]. Indeed, contemporary research shows that socioenviromental discrimination and recent healthcare experiences with medical professionals influence medical mistrust among African American men [39]. More recently, experimental evidence finds a relationship between racial discrimination and medical mistrust, revealing that exposure to racial discrimination significantly increases medical mistrust among African Americans [41]. Regardless of the mechanism—be they historical legacies of mistreatment or modern-day personal experiences with medical professionals—it is clear that scientific trust varies by a person's race or ethnicity.

The COVID-19 pandemic has once again revealed racial disparities in health outcomes between White and Black Americans. Data on hospitalization rates, infections, and deaths report that people of color comprise a disproportionate share of the human toll wreaked by the pandemic. Indeed, minority communities have had substantially higher fatality rates than

White communities: According to the Kaiser Family Foundation, "people of color represented more than half of all people tested (57%) and confirmed cases (56%) at health centers, and that Hispanic patients made up a higher share of positive tests compared to their share of total tested patients" [42]. Like immunization disparities, these unequal effects reflect larger underlying social and political factors that are fueled by the historical and systemic discrimination against minorities in the United States. Following this pattern, a 2020 Pew Research Center poll indicates that Black adults place lower confidence in scientists than White adults: 27% of Black Americans report having a 'great deal' of trust in Scientists, compared to 44% for White adults [21]. These differences in trust are clearly concerning because when individuals do not trust science, they may be less willing to support policies that scientists formulate.

## Racial disparities in COVID-19 deaths

To further motivate the importance of our research question and proceeding analyses, we first identify, above and beyond previous literature, the significant and widespread racial disparities present in the United States' COVID-19 deaths. Specifically, we test if communities of color had disproportionately higher rates of death due to COVID-19. Using daily death data from *The New York Times* [43] and demographic data from the *American Community Survey* [44], we specify high dimensional regressions modeling a given county's: (1) cumulative death count and (2) daily change in death counts as a function of the racial composition found in the county throughout the course of the pandemic. We specify our cumulative daily death count in the standard deaths per 100,000 residents and the daily change as a 7 day rolling change average of deaths per 100,000 residents. We elect to take the 7 day rolling average of COVID-19 death count changes within a county following standard measurement conventions given variability in death reporting across county administrative units [45, 46]. Indeed, this measurement is congruent with reporting of COVID-19 daily death rate data in the United States.

The unit of observation of this model is a given county mortality rate on a given day of the pandemic from the confirmation of the first case of COVID-19 in Snohomish County, WA on January 21, 2020 until December 31, 2020. We specify three models per dependent variable measuring COVID-19 death counts, for a total of six models with the fully specified model (3) controlling for other salient county demographic variables, clustering standard-errors by county, and using date fixed-effects given the time component in our county-level panel. Specifically, our county-level models control for percentage foreign-born, median age, median income, percent college educated, percent older population 65+, population density per square mile, and total county population. Furthermore, our inclusion of date fixed-effects allows us to assess cross-sectional differences in COVID-19 deaths as a function of a given county's demographic characteristics, mainly race/ethnicity, throughout the course of the first year of the pandemic [47, 48]. Indeed, date fixed-effects compares all counties as a *cross-section* on every given date, and then averages across all dates to determine the effect of differences in racial/ethnic demographics across counties on COVID-19 deaths. This helps eliminate the threat of biased estimates posed by comparing counties early in the pandemic, with low death counts, to counties later in the pandemic (especially during peaks in deaths), with much higher death counts. Lastly, we specify our model with panel-corrected clustered standard errors by county given that we repeatedly observe individual counties over the course of the pandemic [49].

Importantly, while previous research has found significant racial differences in COVID-19 outcomes (infection, hospitalization, and death) cross-sectionally [1–3, 42, 50, 51], a county-level, US-wide panel analysis has, to our knowledge, yet to be published in a peer-reviewed journal. This lack of panel analysis motivates our own, and allows us to confirm even further significant racial disparities in COVID-19 deaths across the country.

**Table 1. County-level high dimensional regression models assessing racial disparities in COVID-19 death, January 21st–December 31st 2020.**

| | Dependent Variable: Σ Cumulative Daily Death Count | | | Dependent Variable: Δ Daily Death Count Changes | | |
|---|---|---|---|---|---|---|
| | Model (1) | Model (2) | Model (3) | Model (4) | Model (5) | Model (6) |
| Percentage African-American | 0.754*** | 0.754*** | 0.745*** | 0.002** | 0.002** | 0.002** |
| | (0.109) | (0.109) | (0.110) | (0.001) | (0.001) | (0.001) |
| Percentage Hispanic | 0.336** | 0.336** | 0.330** | 0.005** | 0.005** | 0.004** |
| | (0.147) | (0.147) | (0.144) | (0.001) | (0.001) | (0.001) |
| Percentage Asian | -0.966*** | -0.966*** | -0.878*** | -0.011*** | -0.011*** | -0.006** |
| | (0.264) | (0.264) | (0.274) | (0.002) | (0.002) | (0.002) |
| Percentage Multi-Race | -0.336* | -0.337* | -0.372* | -0.004** | -0.004** | -0.004** |
| | (0.179) | (0.179) | (0.172) | (0.001) | (0.001) | (0.001) |
| Percentage White | 0.142 | 0.142 | 0.179 | 0.002 | 0.002 | 0.002* |
| | (0.171) | (0.171) | (0.164) | (0.002) | (0.002) | (0.001) |
| Constant | 36.612*** | 36.613*** | 42.435*** | 0.471*** | 0.471*** | 0.601*** |
| | (9.586) | (9.587) | (10.498) | (0.080) | (0.080) | (0.088) |
| Control Variables | - | - | ✓ | - | - | ✓ |
| Date-Fixed Effects | - | ✓ | ✓ | - | ✓ | ✓ |
| $R^2$ | 0.074 | 0.409 | 0.410 | 0.001 | 0.101 | 0.102 |
| Observation $N$ | 1,087,132 | 1,087,132 | 1,087,132 | 1,087,132 | 1,087,132 | 1,087,132 |
| County $N$ | 3,142 | 3,142 | 3,142 | 3,142 | 3,142 | 3,142 |
| Date $N$ | 346 | 346 | 346 | 346 | 346 | 346 |

Data begins with first U.S. confirmed case on January 21, 2020 in Snohomish County, WA.

**COVID-19 Data**: The New York Times from January 21-December 31, 2020 [43].

**Demographic Data**: 2015–2019 American Community Survey Estimates [44].

All models specified with county panel-corrected clustered standard errors.

Models estimated using the `reghdfe Stata` package.

*$\rho < 0.1$;

**$\rho < 0.05$;

***$\rho < 0.01$

Table 1 reports the results of our models assessing racial disparities with respect to COVID-19 death rates at the county level. We find robust evidence across all model specifications that higher Black and Hispanic populations in a given county are associated with significantly higher cumulative daily death counts and higher rates of daily changes to the death count. By contrast, we also find that Asian and multi-racial populations are significantly associated with lower cumulative daily death counts and a *decrease* in daily death county rate changes in a given county. We also find robust null effects between higher percentages of White populations and both the cumulative daily death count and daily death count changes in a county.

While this county-level panel analysis over the first year of the pandemic is conducted based on best-practices, such as time fixed-effects and county-level clustered standard errors, it is still a regression model of county-level data and thus does not provide causal estimation. However, the consistent and large effects with such a high number of observations provides clear evidence that the pandemic has disproportionately affected communities of color. These disparities in death rates underscore the significance of our subsequent analyses on the linkage between race, scientific trust, and social distancing policy support. Indeed, by subsequently finding that scientific trust is associated with an increase in support for government social

distancing policies across racial groups, we suggest that increasing this trust is critical towards mitigating the standing racial inequalities in COVID-19 deaths across communities.

## Scientific trust, race, and social distancing policy support

### Data & measurement

**Scientific trust.** To evaluate whether scientific trust can help raise support for governmental policies critical to containing the COVID-19 pandemic, we rely on the nationally representative sample provided by Pew's `American National Trends Panel Survey` [5]. To address potential concerns of endogeneity regarding the onset of the COVID-19 pandemic and scientific trust, we rely on two panel survey waves. To measure scientific trust, we rely on wave 42 fielded prior to the onset of the pandemic from January 7–21, 2019, to measure our main explanatory variable of interest, latent scientific trust. While some early work suggests that levels of scientific trust may be stable in the United States throughout the course of the pandemic [52, 22], our measure allows us to avoid potential endogeneity between trust and policy support during the early period of the COVID-19 pandemic. To measure this latent variable, we rely on a series of questions designed to tap into the propensity of an individual to trust the scientific process and scientific community. Specifically, we leverage questions measuring the following: (1) confidence that scientists act in the best interest of the public; (2) whether scientists should take an active role in scientific issue policy debates or not; (3) whether public opinion should play an important role in guiding scientific policy issue decisions; (4) whether scientific experts are better suited to make "good" policy decisions about scientific issues relative to "other" people; (5) whether the scientific method produces accurate conclusions independent of the conclusion the researcher wants; (6) whether scientists make judgements based solely on facts or if they are "biased as other people;" (7) the importance of scientific research that has immediate practical applications for society; and (8) the importance of scientific research to advance knowledge, even if there are no immediate benefits for society.

Given that we are seeking to measure an inherent latent variable (scientific trust) using questions with varying scales, we employ an exploratory factor analysis to derive the structure of latent scientific trust. Results of this two-dimensional promax rotation factor analysis can be found in Fig 1 and the specific factor loadings can be found in S3 Table. We find that questions (1–6) map consistently onto the first dimension, with a high Cronbach's $\alpha$ of 0.66, and questions (7–8) map consistently onto the second dimension. This makes sense given that questions (7–8) focus on the benefits/importance of scientific research, whereas questions (1–6) focus on whether individuals trust scientists and whether they should be involved in the policy process. Given that our research question concerns this latter dimension, we choose to use dimension 1, consisting primarily of questions (1–6), as our measure of scientific trust.

We extract our measure of latent scientific trust in our sample (i.e., first dimension factor scores) and also differences in this measure across racial groups. Note that construction of these racial groups rely on reported self-identification in the Pew survey data. Consistent with previous work in public health [31, 33, 34], we find significant differences in the mean values of latent scientific trust between racial groups. Indeed, we find that White and Asian respondents generally exhibit higher levels of scientific trust than Black or Hispanic respondents. S2 Fig presents further evidence of this racial variation from a fully specified regression model showing that Black and Hispanic respondents possess lower levels of predicted latent scientific trust than White respondents. We also find that Asian-Americans do not possess significant differences in predicted trust than White-Americans. This model controls for other predictors of scientific trust, such as partisanship, ideology, income, gender, education, age, and

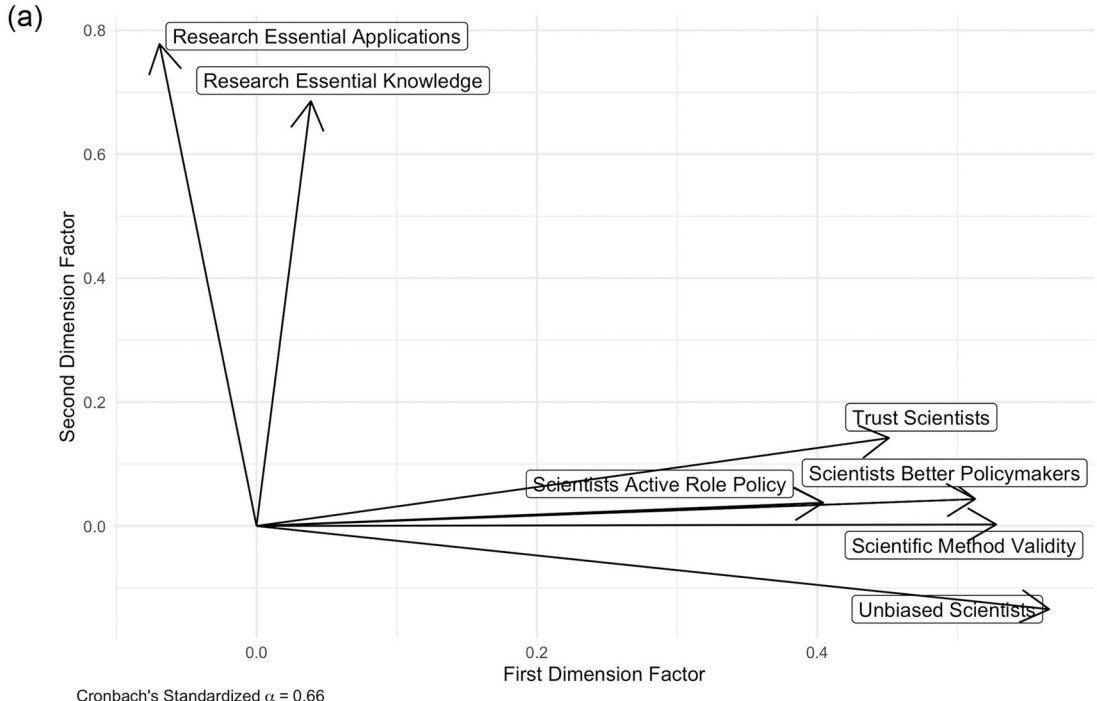

Cronbach's Standardized $\alpha$ = 0.66

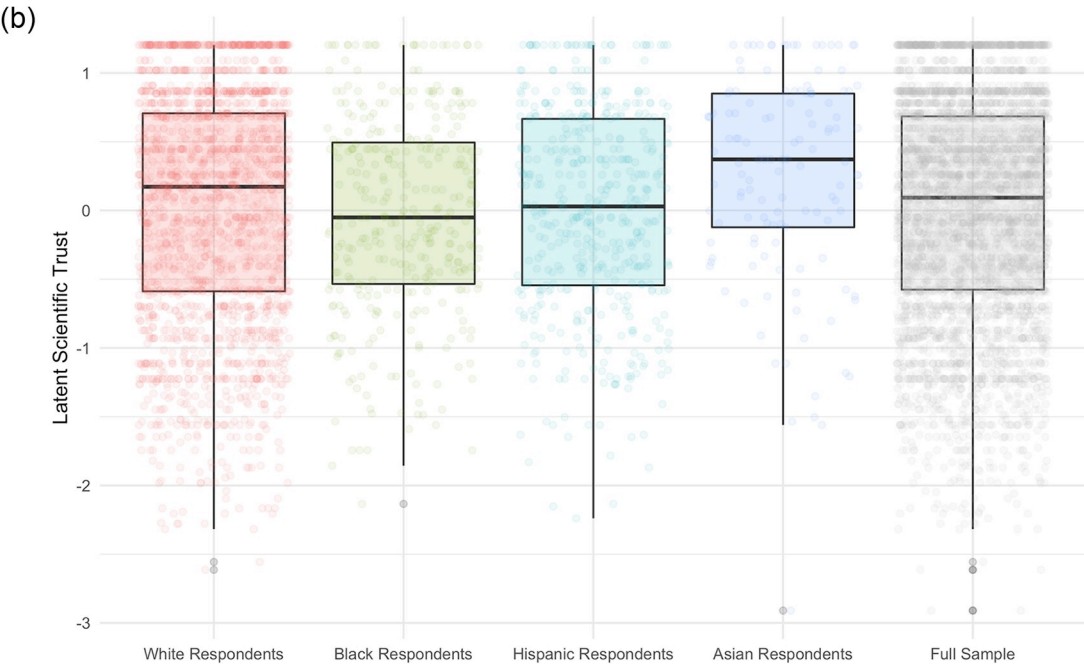

ANOVA suggests significant differences in mean latent scientific trust across racial groups, p < 0.01.

**Fig 1. Measuring latent scientific trust in the mass public.** A: Latent scientific trust as measured by factor analysis. B: Distribution of latent scientific trust by race.

geographic region. Overall, our descriptive finding of the differences in scientific trust between White and Black/Latino respondents adds strong face validity to our measure of latent scientific trust by uncovering a similar distribution across race as the standing literature.

**Social distancing policy support.** To measure our outcome variables of interest, namely support for government social distancing restrictions, we rely on panel wave 64 fielded from March 19–24, 2020. In this survey, panelists were asked:

> Thinking about some steps that have been announced in some areas to address the coronavirus outbreak, in general do you think each of the following have been necessary or unnecessary?

a. Restricting international travel to the U.S.

b. Requiring most businesses other than grocery stores and pharmacies to close

c. Asking people to avoid gathering in groups of more than ten

d. Cancelling major sports and entertainment events

e. Closing K–12 [primary and secondary] schools

f. Limiting restaurants to carry-out only

g. Postponing upcoming state primary elections

Respondents were given two response options, "necessary" and "unnecessary," and the sequence that these items appeared in the list above was randomized. For our purposes, these outcome variables were coded as 1 if people responded that a given step was "necessary" or 0 if they responded that it was "unnecessary."

We construct a composite measure of latent COVID-19 restriction policy preferences using an item-response theory model (IRT), with resulting respondent scores providing a measure of overall preferences for social distancing policies. IRT models are a useful tool for measuring latent preferences or characteristics from a set of observed behaviors, with the canonical example being the measurement of students' abilities with multi-item tests. In the testing example, higher ability should correspond to a higher score from the IRT indicating a higher probability of answering a given question, dependent on that question's own difficulty. In political science, the IRT model has been used to measure ideology, political knowledge, and other latent concepts from a set of observed indicators [53]. In this scenario, using an IRT model makes conceptual sense as each policy can be interpreted as a more or less strict regulation.

As shown in Fig 2, we find that the item characteristic curves for each question map very well with this conception of a dimension of policy options ranging from less strict regulation (further left on the X-axis, e.g., barring international travel) to stricter regulation (further right on the X-axis, e.g., shutting down most businesses). Importantly, these curves should mostly not intersect and should be dispersed along the X-axis. Only restricting international travel and postponing primary elections intersect other curves, which makes some intuitive sense given a) the overwhelming popularity of/bipartisan support for restricting international travel and b) the complicated nature of voting in various states in the U.S., with many states having widespread mail-in voting and others with very limited mail-in voting. Overall, the results of our IRT suggest that using it as a single scale of social distancing policy support is not only

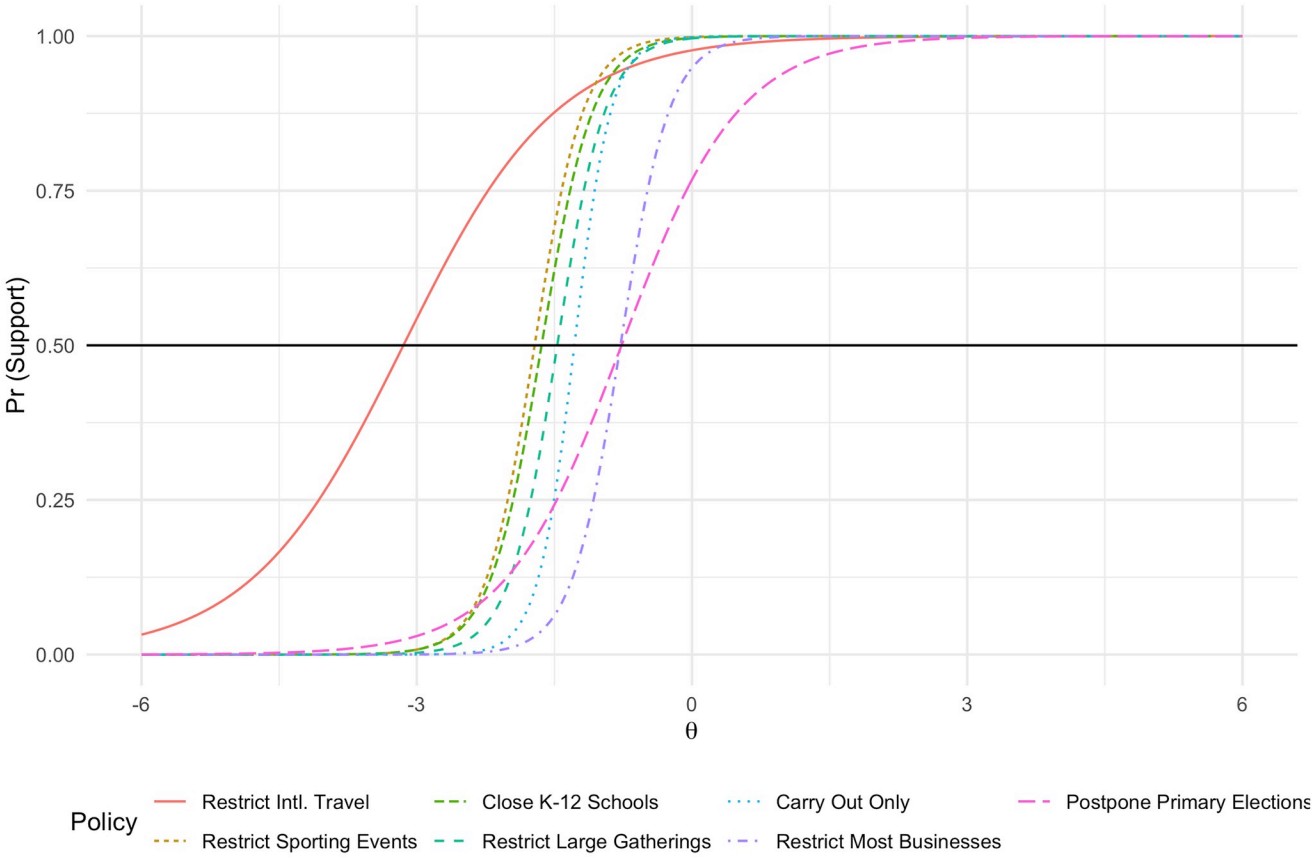

**Fig 2. Item characteristic curves of social distancing policy IRT.**

viable but likely well represents a latent dimension of individual social distancing policy support. Finally, while the IRT scores should only be interpreted relative to one another, they range from -2.4, least supportive of social distancing policies, to 0.553, most supportive of social distancing policies (see S1 Table for descriptive information on all of our dependent variables).

Consistent with our expectations that these policies are determined more by scientific trust, rather than ideology or partisanship, COVID-19 restriction attitudes and the respondent's liberal-conservative identification are only weakly correlated ($\rho = 0.18$), with self-identified liberals slightly more likely to support restrictions. Lastly, we also construct a summated rating scale to evaluate the relationship between scientific trust and degree of policy support as an alternative-measure/robustness-check of aggregate policy support. These scores range from 0, no policies supported, to 7, all policies supported.

Critical to note, one possible problem with surveys that ask respondents for either their policy support for or compliance with COVID-19 policies is social desirability bias. Recent work by Larsen et al. [54] suggests that, at least in Denmark and when asked about compliance, there is no evidence that respondents are under-reporting said compliance. However, in contradiction to this work, three other pieces of experimental research suggest significant under-reporting of compliance with social distancing policies. First, Daoust et al. [55], use a face-saving strategy embedded in three survey experiments of (in total) nearly 6,000 Canadians, that all implement some form of face-saving question wordings and/or response options related to social distancing compliance. Allowing individuals to answer in the negative in a relatively

less-costly way, at least in relation to social costs, leads to a significantly lower self-reported social distancing compliance rate. Second, Daoust et al. [56] apply a similar "guilt-free" treatment to 12 different countries, including the United States, and find that respondents significantly over-report their social distancing compliance in response to direct questions. Importantly, this study expands its dataset to include the United States, which has significantly more relevance to our data and analysis. They find that there are no significant differences in social desirability across age, gender, or education, but do not analyze whether there may be differences across races/ethnicities. Finally, Timmons et al. [57] utilizes a list experiment of online respondents that finds that, when compared to direct questions, list responses lead to a significantly lower percentage of respondents reporting social distance compliance. This would suggest that direct questions regarding individual compliance may lead to over-reporting, and there were significant differences between young/old and rural/urban respondents in levels of compliance. However, race/ethnicity was not analyzed as a subgroup.

Overall, the literature seems to indicate that social desirability bias is a significant threat to self-reported compliance in response to direct questions, such as those we use in our analysis. However, these studies importantly do not examine whether there are significant differences in social desirability bias between racial or ethnic groups in their self-reported (non-)compliance with social distancing policies. Additionally, to our knowledge, no study to date assesses whether there are racial differences in social desirability bias related to social distancing compliance (or similar government orders). A lack of racial/ethnic differences is critical for our findings, as different levels of social desirability bias could create the illusion of policy differences between races/ethnicities. For example, if white respondents were more prone to social desirability bias than black respondents, but in truth white and black respondents had the same level of policy support, we would then erroneously find significant differences between white and black respondents.

Furthermore, while these findings are not perfectly applicable to our work in the United States on *policy support* (*not compliance)*, other factors such as the lack of consistent policies across states [58] and the partisan nature of support for and compliance with social distancing policies [26, 59] reduces the threat of social desirability bias among respondents. First, given that we are focusing on policy support, not actual compliance, the threat should be lower given that risky behavior is more likely to be shamed than simple opposition to a policy. Importantly as well, we have significant variation across our measures of policy support within our salient demographics, namely race/ethnicity and partisanship. If social desirability bias had a consistent effect on all respondents, which some of the literature finds, then our estimates would be conservative in nature as all respondents would have elevated levels of support (because the question has a binary response, respondents would appear exactly the same if they were socially pressured to respond with "necessary"). Finally, to reiterate what was said above, we have no reason to believe that a social desirability bias varies by race/ethnicity, which would be necessary to bias our results in such a way as to result in a false-positive in differences between race/ethnicity. Given all of these considerations, social desirability bias is unlikely to affect our substantive results. That being said, our results should be framed in relation to the lack of literature on racial/ethnic differences in social desirability bias, and these potential differences across races/ethnicities cannot be fully ruled out.

**Media and institutional trust, race, and other covariates.**   To test whether scientific trust is a more salient predictor than media or institutional trust, we specify a series of baseline logistic regression models for each of our individual outcome variables measuring a citizen's support for social distancing policies. Importantly, every model is specified with appropriate survey weights, provided by Pew, which allows the sample to match population benchmarks on the basis of salient demographic characteristics such as age, gender, education, race/

ethnicity, and foreign-born status. Inclusion of these survey weights in our model estimation allow us to account for potential bias produced by survey non-response and sampling error, with these weights assisting in producing estimates that are representative of the general population.

Given our theoretical framework, we expect the marginal effect of scientific trust on the probability of supporting COVID-19 containment public policies to be larger than the other two forms of trust. We measure trust in the media and institutions from wave 42, the same survey wave preceding the pandemic and used to measure latent scientific trust. These two trust variables are measured on a scale of 1 (no confidence at all) to 4 (a great deal) from survey questions asking respondents to indicate their trust in the news media and elected officials (the specific question wording can be found in S1 Appendix). We also specify our baseline model with standard predictors of policy preferences, such as gender, political ideology, age, education, income, race, and geographic region. With regards to race, we specify a series of dichotomous dummy variables to indicate if a respondent identified as African-American, Hispanic-American, and Asian-American with majority White identification being the baseline category. In our sample, approximately 70.91% ($N = 1,855$) of respondents identified as White, 10.97% ($N = 287$) as African-American, 14.83% ($N = 388$) as Hispanic-American, and 3.29% ($N = 86$) as Asian-American. We extract Asian identification from the "other" coding provided by the race-ethnicity variable and additional information provided by a variable that expands on the initial race coding convention. This coding follows and expands standard race coding conventions provided by the Pew Survey.

## Analyses

In terms of our two measures of latent and summated composite COVID-19 social distancing policy support, we specify ordinary least squares (OLS) regression models using the same set of variables as our individual policy logistic regression models. In these OLS models predicting a citizen's latent and summated COVID-19 containment policy support, we similarly expect the marginal effect of scientific trust to be larger than that of media and institutional trust.

Lastly, our theoretical framework posits that scientific trust should increase policy support for social distancing policies, even across racial cleavages in the United States. Indeed, we argue that increased latent scientific trust can increase support of these critical public policies across differing racial communities. We also posit that scientific trust works in differing ways than media and institutional trust, in that scientific trust raises support for these policies across racial categories while media and institutional trust do not. To test this argument, we take our baseline model and include an interaction between an individual's racial identity and latent scientific trust. To fully specify the model, we also interact racial identity with media and institutional trust. This allows us to evaluate the marginal effect of each type of trust across racial categories and compare the magnitude of these effects on support for social distancing policies. The expectation in this specification is that across individual, latent, and summated COVID-19 containment policies, the marginal effect of scientific trust should be positive and significant. By contrast, we expect the marginal effect of media and institutional trust to be smaller in magnitude to scientific trust across racial categories.

## Results

### Baseline trust effects on COVID-19 policy support

We now turn to the results of our analysis. Fig 3 shows the marginal effect of going from the minimum to maximum value of latent scientific trust, media trust, and institutional trust on the probability of individual policy support. With the exception of restricting international

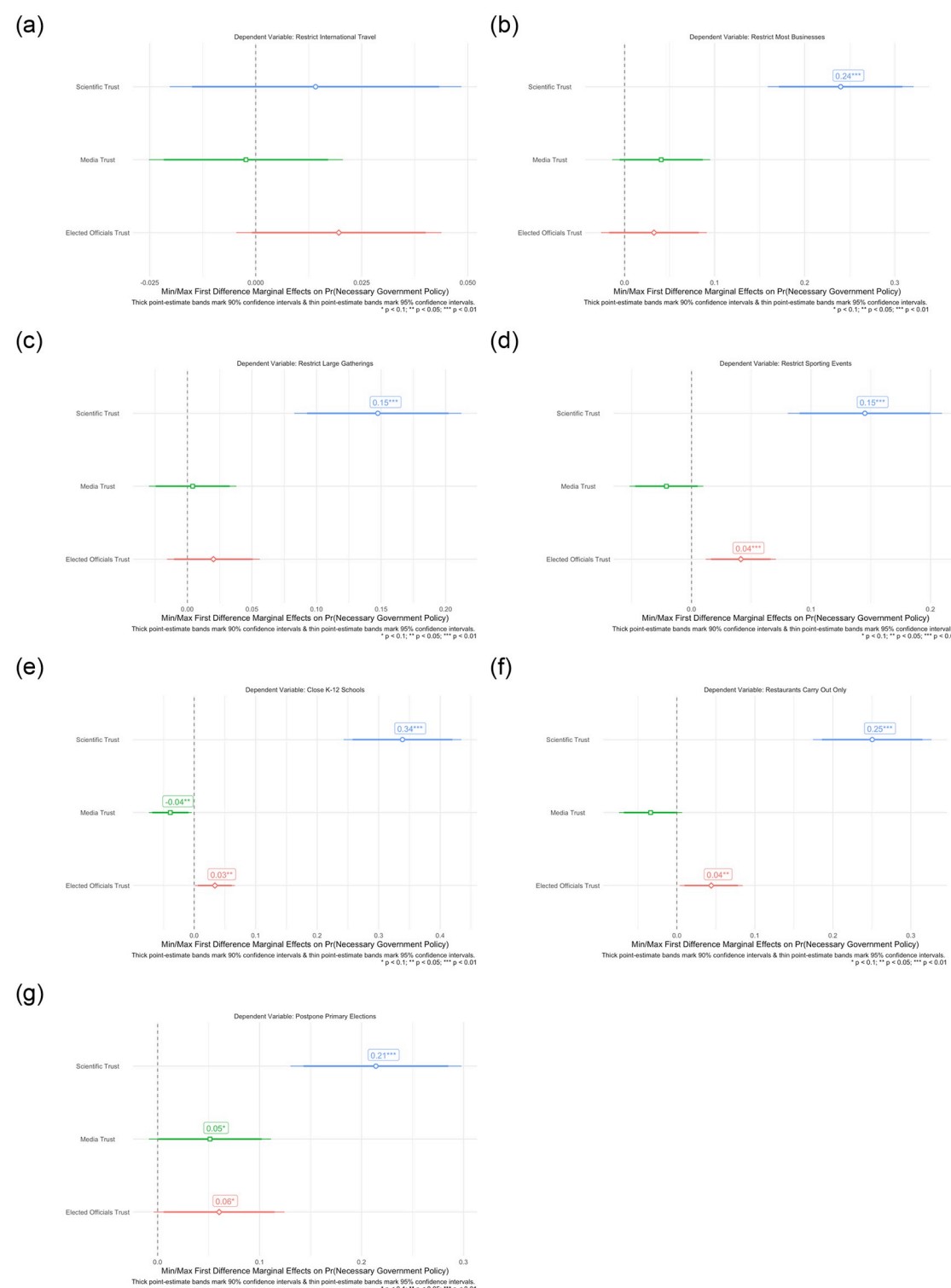

**Fig 3. Baseline model effects of scientific, media, & institutional trust on specific COVID-19 social distancing policy support.**
A: Restrict international travel. B: Close most businesses. C: Restrict large gatherings. D: Restrict major large & sporting events. E: Restrict K–12 schooling. F: Restrict restaurant dining. G: Postpone primary elections.

travel, latent scientific trust is a significant predictor of COVID-19 restriction policy support. Indeed, there is a high degree of agreement in restricting international travel during the onset of COVID-19 in March 2020, with 96.3% of respondents supporting this containment policy. By contrast the other restriction policies attracted only 74.5% (most businesses), 90.6% (large gatherings), 93% (sporting events), 92.2% (K–12 schools), 87.8% (restaurant dining), & 69.5% (postponing primaries). Going from the minimum value of latent scientific trust to the maximum value is associated with an increase in the probability of supporting closing most businesses by 24%, restricting large gatherings by 15%, restricting sporting events by 15%, restricting K–12 schooling by 34%, restricting restaurant dining by 25%, and postponing state primary elections by 21%, respectively. By contrast, our models find a small, significant relationship between greater institutional/elected officials trust and support for restricting sporting events (4%), K–12 schooling (3%), restaurant dining (4%), and postponing primary elections (6%). These associations are minimal in magnitude, ranging from a 3% increase in probability of supporting restrictions to K–12 schooling to a 6% increase in supporting postponing state primary elections. Lastly, increased trust in the media is only associated with a 5% increase in the probability of supporting postponement of primary elections and, in fact, a slight decrease in the probability of restricting K–12 schooling by 4%. In all, we find strong support that latent scientific trust is a far more salient predictor of individual COVID-19 containment policies than other sources of trust after accounting for standing predictors of policy preferences.

We now turn to evaluating our OLS models predicting our measure of latent and summated COVID-19 restriction policy support. Congruent with the findings of individual policies in Figs 3 and 4 shows that latent scientific trust, and institutional/elected official trust, significantly correspond to greater support for overall COVID-19 restriction policies in both our latent and summated policy measures. Once again, the effect of latent scientific trust as a predictor of overall policy support is larger than institutional trust in the media across both OLS models. Indeed, going from the maximum to minimum value of latent scientific trust corresponds to a significant predicted increase of 1.16 in the summated policy support scale. The magnitude of this effect is noteworthy, given that this summated COVID-19 policy support scale is on a scale of 0 to 7. By contrast, this same effect in terms of institutional trust for elected officials corresponds to only an increase of 0.4 in the summated policy scale. The effect of media trust on predicted latent and summated COVID-19 restriction policy support is insignificant across both models. Taken together, and both in terms of individual and summated policies, we find strong support for our baseline expectations that latent scientific trust is not only associated with increases in support of COVID-19 social distancing policies, but is also a more salient predictor of these policy preferences than trust in the media and government institutions/elected officials. For additional information and the tabular results of the regressions illustrated in Figs 3 and 4, see S5 Table.

## The consistent effects of scientific trust across race

Building off our strong findings in the baseline models, we turn to evaluating our interactive models assessing latent scientific trust across racial identification in our sample. We posit that across all racial cleavages, latent scientific trust should raise the probability of supporting individual COVID-19 restriction policies. Fig 5 evaluates this hypothesis from our interactive models. Unlike the two other forms of trust, there is strong evidence that latent scientific trust raises the probability of policy support across racial cleavages. Indeed, greater latent scientific trust corresponds to greater support for all racial cleavages (i.e., Hispanic, White, Black, Asian) for restricting large sporting events, K–12 schooling, and restaurant dining as shown in

(a)

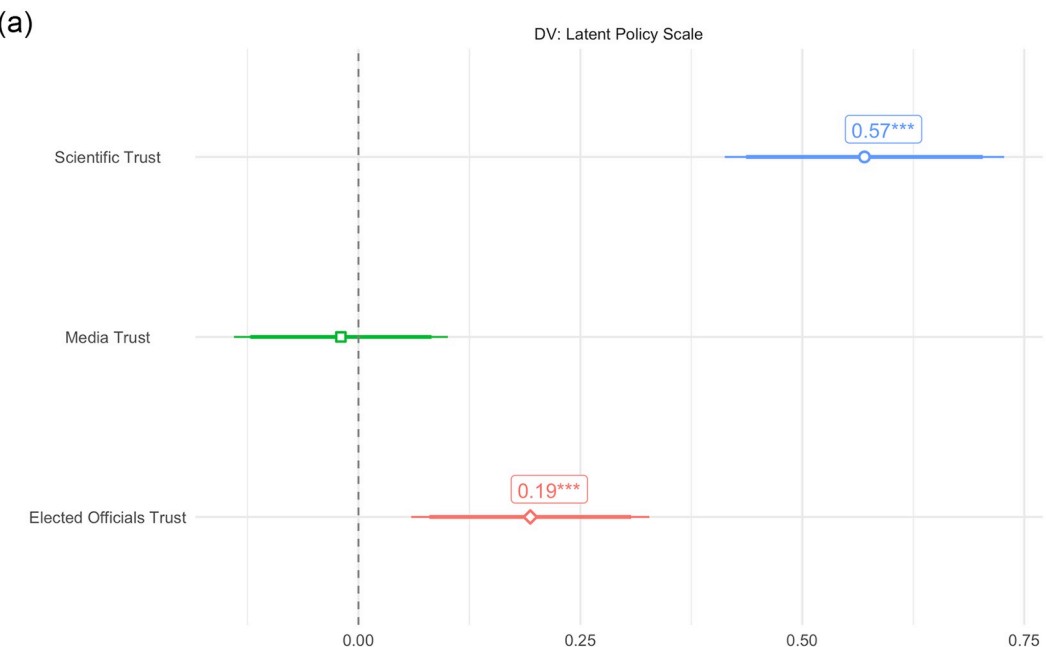

(b)

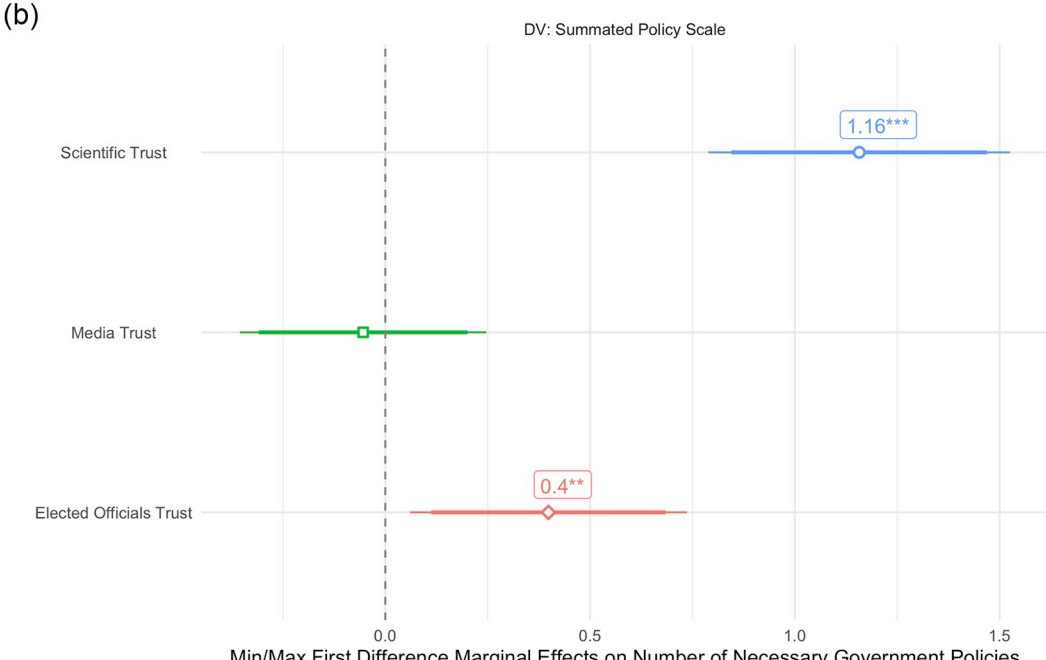

**Fig 4. Baseline OLS model effects of scientific, media, & institutional trust on composite COVID-19 social distancing policy support.** A: Latent policy measure. B: Summated policy support.

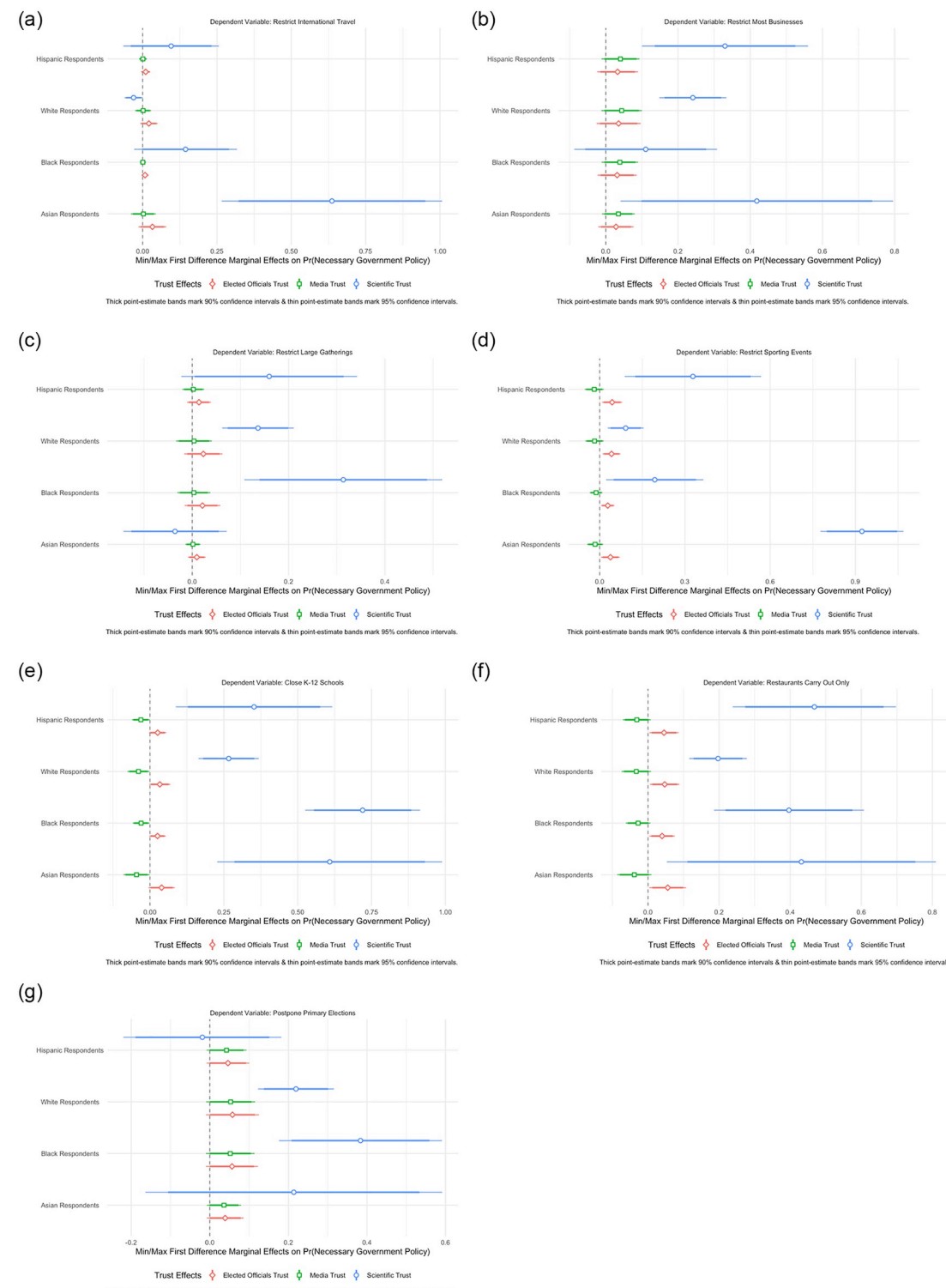

**Fig 5. Measuring latent scientific trust in the mass public.** A: Restrict international travel. B: Close most businesses. C: Restrict large gatherings. D: Restrict major large & sporting events. E: Restrict K–12 schooling. F: Restrict restaurant dining. G: Postpone primary elections.

Fig 5D–5F. In terms of restricting international travel, greater scientist trust only significantly raised the probability of this policy support among Asian-Americans as shown in Fig 5A. Fig 5B shows that this effect was significant for all racial categories with the exception of African-Americans in the context of closing most businesses. Fig 5C and 5G shows only a significant effect for White and African-American respondents in the context of restricting large gatherings and postponing primaries, respectively. Lastly, we find overwhelming evidence of insignificant or minimal trust effects for elected officials and the media, suggesting that these effects do not substantively raise the probability of policy support across races. Across our 7 individual policy models, we find a significant positive effect of latent scientific trust in 4 models for Hispanic-Americans, 6 models for White-Americans, 5 models for African-Americans, and 5 models for Asian-Americans.

Turning to our composite measures of latent and summated COVID-19 policy preferences, Fig 6 finds strong support that latent scientific trust across both of these OLS measures predicting these outcome measures for all racial cleavages. Fig 6B finds that going from the minimum to the maximum level of latent scientific trust raises the predicted value of summated policy support by 2.23, 1.00, 1.68, and 1.33 for Asian, White, African-American, and Hispanic respondents, respectively. Aside from a minimal elected officials trust effect for White Americans in both OLS models, all other forms of trust are insignificant across both the latent and summated policy models. The results of these OLS models provide clear evidence that, in absolute terms, higher levels of latent scientific trust correspond to higher overall policy support for COVID-19 containment policies independent of measuring this support in latent or aggregate terms. For additional information and the tabular results of the regressions illustrated in Figs 5 and 6, see S6–S8 Tables.

## Discussion

COVID-19 has wreaked a profound toll on human life in most of the world, with much of its impact being concentrated unequally among marginalized communities and people of color, as shown in our county-level analysis. Given COVID-19's continuing toll before vaccination and thus immunity is widespread, and the high potential for another pandemic in the future [60], it is critical to understand how an individual's own characteristics and demographics influence their trust in science and thus their willingness to adopt behaviors that comply with scientific-based health policies and mandates. Furthermore, it is even more important to understand the specific interplay between race/ethnicity and scientific trust so as to mitigate, or better yet prevent, future, outsized damage to communities of color caused by a pandemic. Our research aims to aid in this understanding, already being investigated by other scholars [61], by exploring how scientific trust interacts with race/ethnicity to influence support for social distancing policies.

We find that scientific trust not only is associated with increased support across all races, but has particularly large effects among Black respondents and has a larger impact than both trust in media and government institutions/elected officials. This both underscores the importance of scientific trust in determining support for health policies, specifically those relating to social distancing, and points to a clear avenue for future intervention. Specifically, increasing trust in science within communities of color has the potential to significantly increase support for and potentially compliance with social distancing policies specifically, and public health orders and recommendations more generally. Clearly, our results indicate both the need for and high potential return from building inroads between marginalized communities and the scientific community. Depending on the time-frame, a Black-specific intervention could likely help Black Americans' support for COVID-19 related policies and recommendations aimed at

(a)

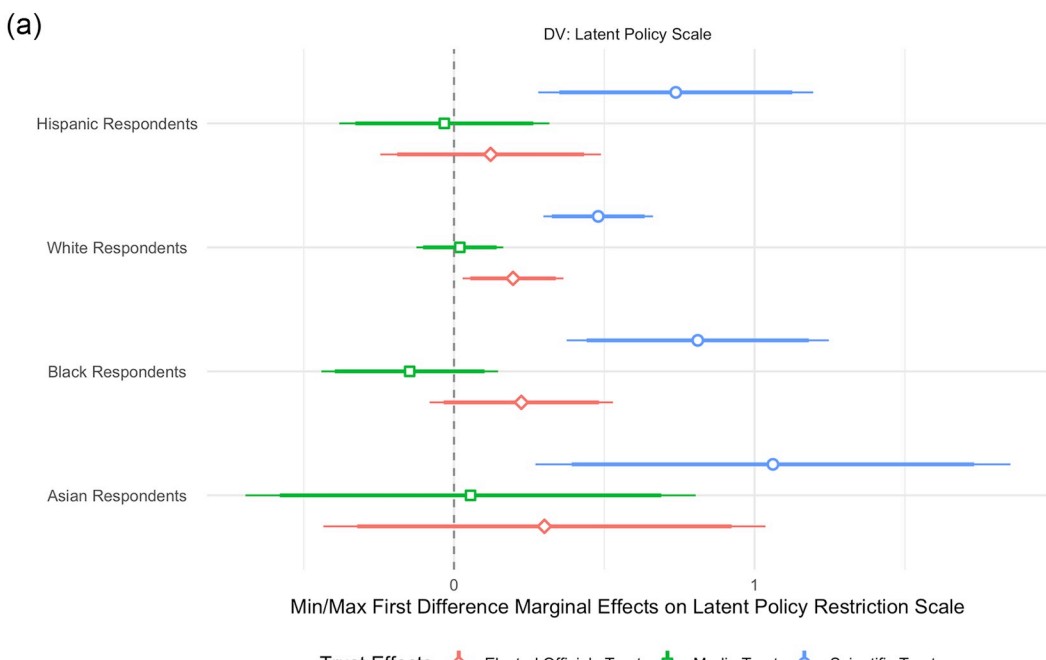

(b)

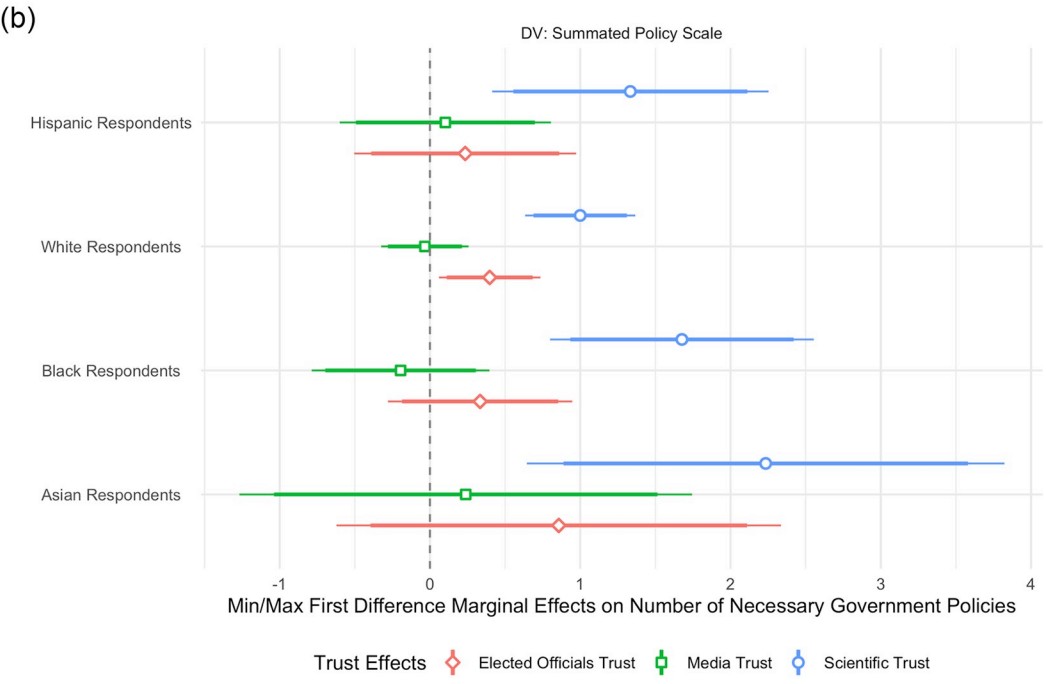

**Fig 6. Baseline OLS model effects of scientific, media, & institutional trust on composite COVID-19 restriction policy support across race.** A: Latent policy measure. B: Summated policy support.

reducing infection rates and thus, mortality. While compliance with and support for social distancing policies is not the only determinant of increased infections and mortality among communities of color in the U.S. (historic legacies of racism influence other determinants such as poverty and a lack of access to medical care), it is likely significant given the previous literature on the yearly influenza vaccination rates among people of color [31, 33, 34].

Moreover, our results suggest that trust in science serves as a critical mechanism that can raise support for government policies to help mitigate the societal collective action problem posed by public health challenges. While scholars find a degree of correlation between individual social distancing compliance and support for policies that mandate social distancing [46, 62], scholars also note that voluntary compliance in social distancing practices is not sufficient enough to halt the spread of the COVID-19 pandemic [4, 10]. Consequently, we argue that public support for social distancing guidelines is critical to increasing the likelihood of government enactment of social distancing policies by office-seeking elected elites. Indeed, our results propose that scientific trust can raise public support for critically needed social distancing policies across various communities to mitigate a public health pandemic that disproportionately affects communities of color. We contend that raising scientific trust is essential towards galvanizing critically needed government action on the COVID-19 pandemic and potential future public health calamities that cannot be mitigated by citizen behavioral compliance alone.

Overall, while our research is limited in its findings, namely that we have only isolated the relationship between scientific trust and policy support, not policy compliance, vaccine adoption, or infection rates, it provides an important basis for future research. Specifically, future research should continue to explore not only the determinants of the racial inequality in COVID-19 infections and deaths, but also explore what policies, including increasing scientific trust, could be used to prevent this inequality from occurring in a future pandemic.

Evidence from political science emphasizes the importance of diversity for political trust. For example, Koch (2019) [63] shows that racial minorities anticipate a racial basis by White political elites when it comes to policy-making. Shared experiences between people and elites —be they scientists or politicians—may help facilitate trust [64] and increase support for policies. Evidence from a field experiment in Oakland, California documents that diversity among healthcare providers improves health outcomes for patients [65]. Specifically, Aslan et al. (2019) [65] randomly assign Black male patients to either Black male doctors, or non-Black male physicians and report that shared race between the patient and the doctor increases the patient's willingness to accept preventative services. Importantly, the study attributes these effects to *trust* and better communication between Black physicians and Black patients. Moving forward, we believe that race is an important factor in understanding the public's willingness to trust scientists, and consequently adopt the policies that they recommend. Finally, these results clearly illustrate the devastating and unequal impact of COVID-19 on people of color. The potential to prevent any future, unnecessary deaths, especially among marginalized communities, should strongly motivate this future research.

## Supporting information

**S1 Appendix. Variable coding schemes and question wordings: Pew data.**
(PDF)

**S1 Table. Descriptive statistics of dependent variables: Pew data.**
(PNG)

**S2 Table. Descriptive statistics of independent variables: Pew data.**
(PNG)

**S3 Table. Two-dimensional factor loadings of scientific trust.**
(PNG)

**S4 Table. Scientific trust factor analysis: Descriptive and reliability statistics.**
(PNG)

**S5 Table. Average marginal effects of scientific, media, and political trust on social distancing policy support.**
(PNG)

**S6 Table. Average marginal effects of scientific trust conditional on race/ethnicity on social distancing policy support.**
(PNG)

**S7 Table. Average marginal effects of media trust conditional on race/ethnicity on social distancing policy support.**
(PNG)

**S8 Table. Average marginal effects of political trust conditional on race/ethnicity on social distancing policy support.**
(PNG)

**S1 Fig. Distribution of summated social distancing policy support by race.**
(PNG)

**S2 Fig. Additive OLS regression model results of latent scientific trust.**
(PNG)

## Author Contributions

**Conceptualization:** Sara Kazemian, Sam Fuller, Carlos Algara.

**Data curation:** Sam Fuller, Carlos Algara.

**Formal analysis:** Sara Kazemian, Sam Fuller, Carlos Algara.

**Investigation:** Sara Kazemian, Sam Fuller, Carlos Algara.

**Methodology:** Sam Fuller, Carlos Algara.

**Validation:** Sam Fuller.

**Visualization:** Carlos Algara.

**Writing – original draft:** Sara Kazemian, Sam Fuller, Carlos Algara.

**Writing – review & editing:** Sara Kazemian, Sam Fuller, Carlos Algara.

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
