## [Decision Letter · Decision Letter 0]

25 Mar 2021

PONE-D-21-06826

The role of race and scientific trust on support for COVID-19 social distancing measures in the United States

PLOS ONE

Dear Dr. Fuller,

Thank you for submitting your manuscript to PLOS ONE. After careful consideration, we feel that it has merit but does not fully meet PLOS ONE’s publication criteria as it currently stands. Therefore, we invite you to submit a revised version of the manuscript that addresses the points raised during the review process.

First, let me be clear: I believe that your manuscript is of high quality and it was a real pleasure to read it. As you will see, the reviewers are also quite positive, but outline many comments. R1 has many very useful suggestions and comments that, if taken into account, would improve the manuscript. I should, however, mention that you can use the word ‘result’ although, following the spirit of R1’s comment, you should be clear that you do not uncover a causal relationship. R2 also provides several useful comments. Among others, I agree that clarifications are needed regarding the different measures/scales used. On top of R2’s points, I should add that the items for ‘policy support’ should reflect the exact wording. I assume that it is currently not the case, notably because of the wording of “>10 people” (which would probably not be written that way in the survey). Still on policy support, you mention that the outcome is dichotomized, but was that the initial scale (i.e. necessary versus unnecessary) or did you simplify the original scale? Overall,  I invite you to address both reviewers’ comments and I believe that they will allow you to resubmit a stronger manuscript. Below, I also provide my own comments. 

Is policy support the most important outcome to measure or would not it be more useful to consider levels of compliance with public health preventive measures? I assume that compliance it more closely linked to the disproportionate number of infections and deaths among different subgroups of the population. You briefly acknowledge this in the discussion (e.g. “Specifically, increasing trust in science within communities of color has the potential to significantly increase support for and potentially compliance with social distancing

policies specifically”), but it would benefit from more details. Put it simply: do you rely on the assumption for policy support is a very strong predictor of compliance? 

As for citizens’ compliance with preventive measures, social desirability bias that might at play. The literature is mixed. For example, research using list experiments like Larsen et al. (2020) suggests that it is not an issue, while other research who used a face-saving (or ‘guilt-free’) find evidence or such a bias. Your manuscript would benefit from addressing this potential issue, and most importantly whether it would only affect the distributions of your outcomes (on policy support) or also your inferences. 

You include median age in your model, as well as the percentage of 65+ years old. It is not clear why you need both, and I was wondering if there was a rationale behind it that was currently missing. 

Clarification: why is the N for the observation level varying across the different DVs (Table 1)? 

Given the size of the coefficients, Models 3-6 from Table 1 should have more than two decimals (or you could change the scale of the IV, but you might prefer to simply show more decimals). For example, the effect for White is always 0.00 (0.00) but is said to be statistically significant in model 6. Related to comment #17 of R1, it provides a misleading understanding of statistical significance.

The following work would likely benefit your discussion on the notion of trust: Devine, D., Gaskell, J., Jennings, W., & Stoker, G. (2020).  Trust and the Coronavirus Pandemic: What are the Consequences of and for Trust? An Early Review of the Literature. *Political Studies Review*, 1478929920948684.

Clarification on line 196: “also differences in this measure across racial groups.” Sounds like *the factor loading* is different based on subsamples, but then we understand that you simply compare the means of this measure across these subsamples. It could be clearer right away. 

Clarification needed, line 243: To explain the “appropriate survey weights.”

The Figures should include a note explaining what the CI reflect. I see that there are 2 levels. Are they, for example, 90 and 95% CI? To clarify.

We look forward to receiving your revised manuscript.

Kind regards,

Jean-François Daoust

Academic Editor

PLOS ONE

Journal Requirements:

Reviewers' comments:

Reviewer's Responses to Questions

**Comments to the Author**

1. Is the manuscript technically sound, and do the data support the conclusions?

Reviewer #1: Yes

Reviewer #2: Partly

2. Has the statistical analysis been performed appropriately and rigorously? 

Reviewer #1: Yes

Reviewer #2: I Don't Know

3. Have the authors made all data underlying the findings in their manuscript fully available?

Reviewer #1: Yes

Reviewer #2: Yes

4. Is the manuscript presented in an intelligible fashion and written in standard English?

Reviewer #1: Yes

Reviewer #2: Yes

5. Review Comments to the Author

Reviewer #1: I liked this paper and felt it was useful, well written, and well executed. I nevertheless have some areas for improvement to suggest:

The authors need to make the historical and current context for ethnic minorities come earlier in the paper and be sprinkled more throughout. Otherwise this comes across as a bit deficit framed, i.e. lack of trust rather than situating the trust in historical racism and current racism.

Sometimes this also comes across as a little naïve e.g. p 3 line 87, when the authors start on the racial disparities -there is a huge body of work on racism in science, govt, medicine that needs to be drawn on.

Sometimes the authors also refer to historic racism, when again, it is clear racism is still an issue. This should be framed as historic and current.

Table 1 - these by county analyses could be more firmly linked in - why are they needed? don't we know already there are these disparities? Could be woven in a bit more and more mention made of the need for these analyses.

Where are the survey questions drawn from? Who made them up? Are they a scale? Has anyone done validation work? Has anyone done cross cultural validation? What about measurement equivalence work?

How was race measured? What about mixed race people, what do they count as?

At times this work read a bit USA-centric (also, make it clear what K-12 is please). Similarly, I was wondering why the authors use the term "citizens" - do they mean literal citizens? Residents? Why not just people?

Is there an indicator of assets the authors could draw on? They use income as an indicator of SES but given historical events, some of these groups have had less of a chance to accumulate wealth and ideally that would be controlled for etc.

More care could have been taken around figure quality and labelling. Please pay a bit more attention to detail and revise.

Is there more info about the sample? Even just a few sentences would help as not all readers are American or know much about this panel.

I would encourage the authors to expand a bit more re policy implciations and potential interventions. What have others suggested here?

Reviewer #2: I would like to commend the authors on producing an interesting and important manuscript. I have a number of questions and comments about the manuscript. I hope these are not seen as burdensome or pedantic. My approach to reviewing is to treat the manuscript as though it was given to me by a close colleague who is interested in producing the strongest work possible, and my suggestions are given in that spirit. Especially when there are both multidisciplinary readers and reviewers, some things that may seem "obvious" to the authors may need additional explanation to be parsed by experts from other fields.

Introduction

1. P1, L7: Can you clarify the statement about essential workers? The surrounding sentences exclusively address racial and ethnic minorities. It is unclear whether the authors have (or cite) evidence that racial and ethnic minorities are more likely to be classified as essential workers (thus there is a connection), whether this is an assumption, or whether this is just a helpful aside (in which case, perhaps some restructuring would be useful).

2. P1, LL9-10: Are the data from DC linked to a specific reference? If so, please cite, and if not, please add a reference.

3. P1, LL10-13: Two questions here. First, I think you need to more slowly walk the reader through the logic here. Can you explain in another sentence or two why the higher COVID-19 death rate is consistent with a lack of trust in immunization programs, given that for most of the pandemic vaccinations were not yet available? Second, "trustworthy" implies being worthy of trust, so I assume you mean that they tend to "find" medicine to be less trustworthy?

4. P2, LL17-18: Is it accurate to focus exclusively on anti-lockdown protests? What about anti-mask protests and activities (of which there is a surprisingly high prevalence as well)?

5. P2, LL25-26: I am concerned that this statement may unintentionally be misleading. The cited study indicating increased public trust in science (Sibley et al) was conducted in New Zealand, which is sufficiently different than many other countries' that it may be difficult to generalize even to the US. Several studies sampling from the US have found that trust in science neither increased nor decreased during the pandemic [1-2]. Another study examining epidemics from 1970 to 2018 found that the general international trend is a decrease in trust in science post epidemic exposure [3]. A Dutch study found something more in line with New Zealand (small increase in trust) [4]

6. P2, LL27-29: I am aware of at least 1 study that examined this question, though it is currently a preprint [5].

7. P2, LL36-37: This was also found in Agley [1], except that it was linked to liberal/conservative orientation rather than political party (conservatives generally had lower trust in science).

8. P2, LL48-49: The use of parentheses in this sentence is a little confusing; the reader may confusedly assume that you are indicating that conservative media is liberal when you write "conservative (liberal) media" - since we often use the formulation "topic or phrase (restatement another way)."

9. P3, LL74-77: An article from a few years ago [6] argued for a more multifaceted conceptualization of trust in science. Though it may not be necessary to unpack everything in that article (or even to agree with it), I also think there may be more than confidence at play with this variable.

10. P3, LL97-99: Has any study examined whether Tuskeegee is an underpinning factor? I agree that it makes conceptual sense, but one might hesitate to assert that as a core/primary example without evidence (especially in light of other issues related to discrimination).

11. P4, LL153-155: The use of the word "result" here suggests causality (that is, that higher percentages of Black and Hispanic populations caused higher death counts/rates of daily change). However, it does not seem that the model is designed to permit causal inference.

12. P5, Table 1: The finding for Asians in the first daily death count model is markedly different than in models 2 and 3. Do you have any sense of why this might be the case? It is a little unclear whether the only difference between models 1 and 2 is the lagged DV. If so, why would that have such a differential effect?

13. P5, LL161-165: I understand that this is a preliminary analysis designed to set the stage for your main study. At the same time, it is still important to describe limitations and what should - and should not - be intuited based on the findings.

Methods

14. P6, L178: What is meant by the term "scientific elites"?

15. P6, LL170-208: I understand that the factor analysis is not your primary analysis, but I am concerned that the reader may not fully understand the decisions that you made based solely on Figure 1 and the current text. For example, reviewing Supplemental Figure 3 without additional text, it is not clear why you selected a single factor rather than a 2-factor solution. Please strongly consider sharing other information about your decision-making (you likely already have all of it in your output files) to improve interpretability. I find Dr. Neill's work out of Australia to be a very clear approach to this [7].

16. P7, LL223-224: As with the previous comment, I believe that it would be useful to readers if you shared additional details to improve interpretability of the IRT analysis. For readers in medical sciences, especially, who may not be as familiar with IRT models, this clarification (e.g., even something as simple as the range of composite scores being moved from Table S1 to the main text) will help the reader understand your variable.

Results

17. General: While the figures are very helpful visual tools, please consider providing tables with precise outcomes and confidence intervals, as well as precise p values rather than cutoff ranges (e.g., see the American Statistical Association Statement on p-values) [8].

18. P10, LL346-348: There are quite a number of different models being run with both a composite policy score and individual policy scores. One reason for asking for additional prevision (as in my prior comment) is that it is not fully clear how many models were run, which is especially important given the potential for inflated risk of Type I error with multiple comparisons within the same dataset. This is not to say that you need to apply a blanket correction (e.g., Bonferroni), but it is to say that additional information is needed to assist the reader in reviewing and interpreting your findings.

Discussion

19. P10, LL374-376: You indicate that scientific trust "increases support," implying causality, but it is not clear that your study allows for this inference. It may be more accurate to say that latent scientific trust is associated with support.

20. P10, L376: It may be a bit strong to state that this study itself "confirms" the importance of scientific trust. A more cautious interpretation would be that it provides further robust support for the hypothesis that...

21. P11, LL380: While I agree with your statement that increasing trust in science has the potential to increase support for social distancing policies, there is a substantive gap between support for a policy and compliance with that policy. This study would not seem to provide support for the latter, though it makes sense to suggest it as a future direction for research.

22. P11, LL383-385: Can you clarify what you mean by a "Black-specific intervention"? This study examines support for broad societal-level policies, but this language here implies that increasing trust in science among a specific racial group would actively reduce infection rates and mortality. That is a fairly substantial leap that I am not sure can be justified.

23. P11, LL391-400: Another limitation, as you yourselves point out, is that you measured trust in early 2019, and policy support in March 2020. While some research suggests that trust in science may not have changed during that time, especially in the US [1-2], international data suggest different trends [3-4].

[1] Agley, 2020. https://doi.org/10.1016/j.puhe.2020.05.004

[2] Lina, Bering, & Halberstadt, 2021. https://doi.org/10.1016/j.puhip.2021.100103

[3] Eichengreen, Aksoy, & Saka, 2021. https://doi.org/10.1016/j.jpubeco.2020.104343

[4] Groeniger et al., 2021. https://doi.org/10.1016/j.socscimed.2021.113819

[5] Sulik et al., 2021. https://doi.org/10.31234/osf.io/edw47

[6] Nadelson et al., 2014. https://doi.org/10.1111/ssm.12051

[7] Neill, 2008. https://www.academia.edu/download/30778971/Neill2008_WritingUpAFactorAnalysis.pdf

[8] Wasserstein & Lazar, 2016. https://doi.org/10.1080/00031305.2016.1154108

6. PLOS authors have the option to publish the peer review history of their article (what does this mean?). If published, this will include your full peer review and any attached files.

Reviewer #1: No

Reviewer #2: No

---

## [Author Response · Author response to Decision Letter 0]

5 May 2021

Please see the new Cover Letter (Response to Reviewers.pdf) for detailed responses to all reviewer and editor comments.

---

## [Decision Letter · Decision Letter 1]

8 Jun 2021

PONE-D-21-06826R1

The role of race and scientific trust on support for COVID-19 social distancing measures in the United States

PLOS ONE

Dear Dr. Fuller,

Thank you for submitting your manuscript to PLOS ONE. After careful consideration, we feel that it has merit but does not fully meet PLOS ONE’s publication criteria as it currently stands. Therefore, we invite you to submit a revised version of the manuscript that addresses the points raised during the review process.

Thank you for your very useful response letter to my comments and those from the reviewers. As you will see, they were both satisfied. The actual decision is a minor revision in order to allow you to make some minor changes, but you are on the road to the acceptance. In revising your manuscrit, please conducer the two things below:

First, R2 has three remaining (minor) considerations that you could integrate in a straightforward fashion. 

Second, I would like to go back to my comment about social desirability biases. As I mentioned, the literature is mixed. This is tricky for me because I do not want to nudge you to cite my own work on this topic, but your discussion as of now is very one-sided. It makes use of Larsen et al., which is fine, but does not adequately reflect well the state of the literature and it’s contrasting findings / nuances. For example, Daoust et al. (2020, JEPS) made use of a face-saving strategy embedded in a survey experiment and found quite high levels of social desirability bias. Timmons et al. (2020, BMJ, JECH) made use of a very different technique, i.e., list experiment, but still found high levels of social desirability bias. Moreover, on the key point that its effect, if any, should not be heterogenous across races, Daoust et al. (2021, PLOS One) showed that the impact of social desirability is very similar across major sociodemographic variables. Again, I do not wish to push you toward certain work more than others --- the substantial point is that, as of now, your discussion is too one-sided. 

We look forward to receiving your revised manuscript.

Kind regards,

Jean-François Daoust

Academic Editor

PLOS ONE

Journal Requirements:

Reviewers' comments:

Reviewer's Responses to Questions

**Comments to the Author**

1. If the authors have adequately addressed your comments raised in a previous round of review and you feel that this manuscript is now acceptable for publication, you may indicate that here to bypass the “Comments to the Author” section, enter your conflict of interest statement in the “Confidential to Editor” section, and submit your "Accept" recommendation.

Reviewer #1: All comments have been addressed

Reviewer #2: All comments have been addressed

2. Is the manuscript technically sound, and do the data support the conclusions?

Reviewer #1: Yes

Reviewer #2: Yes

3. Has the statistical analysis been performed appropriately and rigorously? 

Reviewer #1: Yes

Reviewer #2: Yes

4. Have the authors made all data underlying the findings in their manuscript fully available?

Reviewer #1: Yes

Reviewer #2: Yes

5. Is the manuscript presented in an intelligible fashion and written in standard English?

Reviewer #1: Yes

Reviewer #2: Yes

6. Review Comments to the Author

Reviewer #1: The authors have addressed all of the comments well. Thank you for responding so constructively. Best wishes.

Reviewer #2: I would like to thank the authors for their attention to detail in revising this manuscript. This is an important piece of work and I look forward to seeing it published. I have no concerns regarding the manner in which you responded to all of the prior suggestions. In completing a final readthrough, I have a few minor suggestions, but, in my opinion, these do not rise to the level of necessitating an additional round of peer review.

(All pagination is for the clean version of the article)

P1, Abstract: The word "communities" is duplicated in the first sentence. In addition, either "trust" or "institutional trust" should be removed from the second sentence to avoid duplication.

P2, L14: "much less trusting" than whom?

P9, L365: Do you mean "unlikely to affect our results substantively" rather than "unlike to affect our substantive results"?

7. PLOS authors have the option to publish the peer review history of their article (what does this mean?). If published, this will include your full peer review and any attached files.

Reviewer #1: No

Reviewer #2: No

---

## [Decision Letter · Decision Letter 2]

21 Jun 2021

The role of race and scientific trust on support for COVID-19 social distancing measures in the United States

PONE-D-21-06826R2

Dear Dr. Fuller,

We’re pleased to inform you that your manuscript has been judged scientifically suitable for publication and will be formally accepted for publication once it meets all outstanding technical requirements.

Kind regards,

Jean-François Daoust

Academic Editor

PLOS ONE

Additional Editor Comments (optional):

Congratulations!

Reviewers' comments:

Reviewer's Responses to Questions

**Comments to the Author**

1. If the authors have adequately addressed your comments raised in a previous round of review and you feel that this manuscript is now acceptable for publication, you may indicate that here to bypass the “Comments to the Author” section, enter your conflict of interest statement in the “Confidential to Editor” section, and submit your "Accept" recommendation.

Reviewer #2: All comments have been addressed

2. Is the manuscript technically sound, and do the data support the conclusions?

Reviewer #2: Yes

3. Has the statistical analysis been performed appropriately and rigorously? 

Reviewer #2: Yes

4. Have the authors made all data underlying the findings in their manuscript fully available?

Reviewer #2: Yes

5. Is the manuscript presented in an intelligible fashion and written in standard English?

Reviewer #2: Yes

6. Review Comments to the Author

Reviewer #2: (No Response)

7. PLOS authors have the option to publish the peer review history of their article (what does this mean?). If published, this will include your full peer review and any attached files.

Reviewer #2: No

---

## [Editor Report · Acceptance letter]

28 Jun 2021

PONE-D-21-06826R2 

The role of race and scientific trust on support for COVID-19 social distancing measures in the United States 

Dear Dr. Fuller:

I'm pleased to inform you that your manuscript has been deemed suitable for publication in PLOS ONE. Congratulations! Your manuscript is now with our production department. 

Kind regards, 

on behalf of

Dr. Jean-François Daoust 

Academic Editor

PLOS ONE